# Severity Outcomes among Adult Patients with Primary Immunodeficiency and COVID-19 Seen in Emergency Departments, United States, April 2020–August 2021

**DOI:** 10.3390/jcm12103516

**Published:** 2023-05-17

**Authors:** Emily Drzymalla, Ramal Moonesinghe, Katherine Kolor, Muin J. Khoury, Lyna Schieber, Adi V. Gundlapalli

**Affiliations:** 1Office of Genomics and Precision Public Health, Office of Science, Centers for Disease Control and Prevention, Atlanta, GA 30333, USA; 2Division of Overdose Prevention, National Center for Injury Prevention and Control, Centers for Disease Control and Prevention, Atlanta, GA 30333, USA; 3The Center for Surveillance, Epidemiology, and Laboratory Services, Office of the Director, Centers for Disease Control and Prevention, Atlanta, GA 30333, USA

**Keywords:** COVID-19, primary immunodeficiency, SARS-CoV-2, public health, inborn errors of immunity

## Abstract

Primary immunodeficiencies (PIs) are a group of diseases that increase susceptibility to infectious diseases. Few studies have examined the relationship between PI and COVID-19 outcomes. In this study, we used Premier Healthcare Database, which contains information on inpatient discharges, to analyze COVID-19 outcomes among 853 adult PI and 1,197,430 non-PI patients who visited the emergency department. Hospitalization, intensive care unit (ICU) admission, invasive mechanical ventilation (IMV), and death had higher odds in PI patients than in non-PI patients (hospitalization aOR: 2.36, 95% CI: 1.87–2.98; ICU admission aOR: 1.53, 95% CI: 1.19–1.96; IMV aOR: 1.41, 95% CI: 1.15–1.72; death aOR: 1.37, 95% CI: 1.08–1.74), and PI patients spent on average 1.91 more days in the hospital than non-PI patients when adjusted for age, sex, race/ethnicity, and chronic conditions associated with severe COVID-19. Of the largest four PI groups, selective deficiency of the immunoglobulin G subclass had the highest hospitalization frequency (75.2%). This large study of United States PI patients provides real-world evidence that PI is a risk factor for adverse COVID-19 outcomes.

## 1. Introduction

An important component of COVID-19 research has been investigating risk factors for COVID-19 (the disease caused by the SARS-CoV-2 virus) and adverse COVID-19 outcomes. During the pandemic, research has provided insights into the associations between chronic diseases and adverse COVID-19 outcomes. Chronic diseases such as heart disease, chronic kidney disease, cancer, diabetes, and chronic obstructive pulmonary disease have been found to increase the risk of severe illness [1]. Primary immunodeficiencies (PIs) are another important group of diseases that may increase the risk of severe illness due to COVID-19 [2]. PIs are inherited defects in the immune system that increase susceptibility to infectious diseases and the severity of outcomes [3]. There are over 400 different types of PIs [4]. These different types affect different parts of the immune system and, as a result, have varying symptoms and severity.

Previous studies, including case reports and cohort studies, have investigated the association between PI and COVID-19 outcomes. These studies found varying, but generally higher, rates of infection, hospitalization, intensive care unit (ICU) admission, invasive mechanical ventilation (IMV), and mortality in patients with PI [5,6,7,8,9,10,11,12,13,14,15,16]. The few studies that directly compared COVID-19 outcomes between patients with and without PI generally found similar rates of severe COVID-19 outcomes between the two groups [17,18,19]. However, the interpretation of these findings is challenging due to study limitations such as sample sizes (1–169 patients) and the overrepresentation of PI patients with severe COVID-19 outcomes due to recruiting participants from hospitals [5,6,7,8,9,10,11,12,13,14,15,16,17,18,19]. In order to further explore the effect of COVID-19 on PI patients and address the limitations of small sample size of earlier studies, we used Premier Healthcare Database (PHD) to look at COVID-19 outcomes in PI patients compared with non-PI patients.

## 2. Materials and Methods

This study was approved by the Centers for Disease Control and Prevention and was deemed exempt from institutional review board oversight as per 45 CFR §46.101 (b) (4) and exempt from patient informed consent as per 45 CFR §164.506 (d) (2) (ii) (B) because the disclosed PHD-SR data had already been deidentified. This report followed the Strengthening the Reporting of Observational Studies in Epidemiology (STROBE) guidelines [20].

### 2.1. Data Source and Study Population

PHD is a large, U.S. hospital-based, service-level, all-payer database that contains information on inpatient discharges, primarily from geographically diverse, non-profit, nongovernmental, and community and teaching hospitals and health systems from rural and urban areas. Inpatient admissions include more than 13 million visits per year, representing approximately 25% of annual United States ED and inpatient admissions. Patient demographics and clinical information available in PHD have been validated [21] and extensively used in clinical and scientific health studies [22,23,24]. Records of deaths at home or hospice after hospital discharge are also included in the dataset [21]. We used Premier Healthcare Database Special COVID-19 Release (PHD-SCR; release date: 14 September 2021), which includes all inpatient and outpatient discharges from 970 hospitals.

We included 873 hospitals in the PHD-SR medical facilities that contributed both emergency department (ED) and inpatient encounter data. All adult ED patients were classified either as ED only, if they were not subsequently admitted, or as inpatients admitted through the ED. We included patients aged 18 years or older who met all of the following criteria: (1) presence of COVID-19 disease identified with an appropriate COVID-19 diagnostic code using *International Statistical Classification of Diseases and Related Health Problems, Tenth Revision, Clinical Modification (ICD-10-CM)* [25] *code* U07.1 (COVID-19, virus identified) from 1 April 2020 to 31 August 2021 [22]; (2) ED visit for acute illness; (3) admission between 1 January 2020 and 31 August 2021; (4) hospital or outpatient discharge or death occurred between 1 April 2020 and 31 August 2021. Patients with PI were identified using a list of 42 ICD-10 diagnosis codes (Table 1); those with ICD code D71 (“functional disorders of polymorphonuclear neutrophils”, which includes chronic granulomatous disease as the prominent diagnosis) were excluded, as a review of patient demographics assigned this code indicated a large proportion of older adults and females. These 42 ICD-10 codes were obtained by scanning the American Academy of Allergy, Asthma, and Immunology (AAAAI) list for immunodeficiency ICD codes for PI-related ICD-10 codes. This list was supplemented with ICD-10 codes for PIs not on the AAAAI list using Jeffery Modell Foundation (JMF) 2020–2021 Global Survey (F. and V. Modell, *personal communication*) as a reference.

We considered four outcomes for clinical severity: hospitalization, ICU admission, IMV use, and death. These four levels of severity were defined by hospital discharge codes and patient billing codes, as appropriate [21,24,26].

### 2.2. Statistical Analysis

We used PIs as the primary exposure variable and identified PI patients with at least one ICD-10-CM diagnostic code recorded between 1 January 2019 and the patient’s first COVID-19 visit (Table 1). We used descriptive analysis to characterize patients at each level of severity of disease according to whether they had PI or not. Multivariate logistic regression models were used to estimate the strength of the association between PI and each of the four outcomes of clinical severity in COVID-19 patients. Covariates included demographic factors (age, race/ethnicity, and sex) and the presence of chronic comorbidities known to increase the risk of the severity of COVID-19 disease. These included cancer, chronic kidney disease (CKD), chronic obstructive pulmonary disease (COPD), obesity, type 2 diabetes mellitus, and cardiovascular disease (heart failure, coronary artery disease, and cardiac dysrhythmia) [27]. Using Healthcare Cost and Utilization Project Elixhauser Comorbidity Software with R package icd, these comorbidities were identified using ICD-10 codes recorded between 1 January 2019 and the patient’s first COVID-19 visit [28,29]. We also used a multivariate regression model using the same covariates to estimate the difference in the mean length of stay at the hospital between PI patients and non-PI patients. Chi-squared tests were used to detect significant differences in characteristics between PI and non-PI patients. SAS (version 9.4; SAS Institute, Cary, NC, USA) was used to conduct statistical analyses that considered clustering by hospital. *p*-values < 0.05 were considered statistically significant.

## 3. Results

### 3.1. Descriptive Characteristics

Based on the inclusion criteria, the study population from 1 April 2020 to 31 August 2021 consisted of 1,198,283 ED patients, including 536,287 inpatients admitted through the ED. There were 853 COVID-19 patients with PI who were seen in the ED, of which (64.2%) were female (Table 2).

The majority of patients were 55–64 (21.9%), while a minority of patients were 18–39 (16.9%). The majority of these patients were non-Hispanic white (76.1%), and about 45.5% of patients had cardiovascular disease, the most prevalent comorbidity. These patients had 26 out of the 42 possible ICD-10 codes for PI (Table 1).

The most common ICD-10 codes were D83.9 for unspecified common variable immunodeficiency and D80.3 for selective deficiency of the immunoglobulin G subclass, with 568 of 1020 records (56%) (Table 1). Patients may have more than one type of ICD-10 code for PI. There were 1,197,430 COVID-19 patients without PI who were seen in the ED (Table 2). Of these patients, 52.4% were female. The age group with the most patients was 18–39 years old (25.5%), and the age group with the fewest patients was 65–74 years old (15.8%). Almost half (48.6%) were non-Hispanic white, and about 27.1% of patients had type 2 diabetes, the most prevalent comorbidity.

### 3.2. COVID-19 Outcomes

After being seen in the ED, from 1 April 2020 to 31 August 2021, 73.2% of PI patients and 44.7% of non-PI patients were hospitalized; in total, 29.3% of hospitalized PI patients and 22.1% of hospitalized non-PI patients were admitted to the ICU; in total, 19.4% of hospitalized PI patients and 13.6% of hospitalized non-PI patients had IMV due to COVID-19; and 17.8% of hospitalized PI patients and 14.3% of non-PI hospitalized patients died (Table 3).

Among PI patients who had a comorbidity of neoplasm, COPD, CKD, obesity, type 2 diabetes, or cardiovascular disease, a higher percentage of patients were hospitalized, admitted to the ICU, had IMV, or died than non-PI patients with these comorbidities, except for PI patients with CKD who were hospitalized. Of the largest four PI groups (common variable immunodeficiency, selective deficiency of immunoglobulin G (IgG) subclass, defects in the complement system, and selective deficiency of immunoglobulin A (IgA)) with 20 or more patients for the outcome, selective deficiency of the immunoglobulin G (IgG) subclass had the highest frequency of hospitalization (75.2%); defects in the complement system had the highest frequency of ICU admission (41.8%) of hospitalized patients; and common variable immunodeficiency had the highest IMV (24.1%) and death (20.4%) of hospitalized patients (Table 4).

PI patients spent on average 1.91 more days (*p* < 0.001) in the hospital than non-PI patients after adjusting for the covariates. Hospitalization, ICU admission, IMV, and death had higher odds in PI patients than in non-PI patients (hospitalization aOR: 2.36, 95% CI: 1.87–2.98; ICU admission aOR: 1.53, 95% CI: 1.19–1.96; IMV aOR: 1.41, 95% CI: 1.15–1.72; death aOR: 1.37, 95% CI: 1.08–1.74) (Table 5).

## 4. Discussion

Severe COVID-19 outcomes, including hospitalization, IMV, ICU admission, and death, were more frequent in PI patients than in non-PI patients seen in emergency departments. These results provide real-world evidence that PI is a risk factor for adverse COVID-19 outcomes. Most of the previous studies that investigated the association between PI and COVID-19 outcomes looked at rates, such as incidence, ICU admission, and case fatality, among PI patients without direct comparison to non-PI patients within the same study. A study in the United Kingdom found that 53.3% of 60 adult PI and COVID-19 cases were hospitalized, the case fatality ratio was 31.6%, and the inpatient mortality ratio was 37.5% [11]. Another study of 16 adult PI patients positive for SARS-CoV-2 in the United States found the case fatality rate to be 25% [7]. However, in 14 studies to date [5,6,7,8,9,10,11,12,13,14,15,17,18,19], COVID-19 outcomes rates ranged from 0% to 91% for hospitalization, from 0% to 36% for ICU admission, from 0% to 36% for IMV, and from 0% to 36% for mortality. The variety of findings may be a result of the small sample size, the overrepresentation of PI patients with severe COVID-19 outcomes, and the different types of PI subtypes in these cohorts.

Some studies, such as our study, compared COVID-19 outcomes between PI patients and non-PI patients. Concerning COVID-19 infection, Milito (2021) found a similar incidence in Italy between all PI patients (4.01 per 100,000) and the general population (5.22 per 100,000) [17]. Delavari (2021) found children PI patients in Iran to be at 1.23 higher risk of COVID-19 infection than the general population [19]. A study, which was a preprint at the time of this manuscript’s preparation, by Armann (2022) in Germany found children with PI to be at 2.7-fold higher risk of ICU admission than children without PI [18]. Similar to our study, this study found participants in hospitals. However, unlike our study, this study focused on children (≤18 years old). Another study, Milito (2021), found a similar incidence fatality rate between all PI patients (3.81%) and non-PI patients (3.28%). However, the study found a greater difference in the incidence fatality rate between adult PI patients (5.10%) and non-PI patients (3.68%) [17]. This study provided information on adults with PI and COVID-19 (>18 years old), similar to our study. However, patients were ascertained using the Italian registry for primary immunodeficiencies instead of hospital visits.

Our study did not examine the potential impact of COVID-19 vaccination on COVID-19 outcomes among PI patients. The study period was limited to a time frame before widespread COVID-19 vaccination was available. We conducted a preliminary follow-up analysis using data from 14 September 2021 to 16 August 2022 using the same model and found some differences from the odds ratios reported in this paper (unpublished results). Vaccination status may not be reliably represented in PHD due to the COVID-19 vaccination variable being populated from hospital data and not from pharmacies, clinics not participating in PHD, or health departments. In the absence of a reliable marker of vaccination status, the interpretation of these findings was unclear, and they were not further investigated.

A strength of this study is the sample size. This analysis included 853 PI patients during the first time period, which, to our knowledge, is the largest sample size of PI patients with COVID-19. This large sample size allowed the odds ratios to be adjusted for chronic conditions associated with severe COVID-19. However, it is important to point out that the majority of PI patients (76.1%) were non-Hispanic white, while 9.3% were non-Hispanic black, and 8.8% were Hispanic. The underrepresentation of minority populations is likely due to an overall disparity in PI diagnosis that may be due to differences in patient referral and access to care among these groups [30,31,32,33].

We acknowledge several limitations. Patients with PI and COVID-19 that are seen in the ED may be hospitalized as a precautionary measure even when less severely affected, and this could minimize the differences observed. While we identified patients with the most common PI diagnoses, such as those involving humoral immunity (common variable immune deficiency and hypogammaglobulinemia), including patients with ICD-10 code D71 did not significantly affect the results nor the overall trend of severe outcomes noted. Another limitation is the use of ICD-10 codes to identify PI cases. The accuracy of ICD-10 codes varies among conditions [34,35]. To our knowledge, the sensitivity and specific of the ICD-10 codes used in this study are not known. As a result, there may be misclassification of PIs in the study population. Patients with HIV or malignancies were not excluded from the analysis. Patients with PI may have conditions that may exacerbate their immune deficiency, such as malignancies or autoimmune conditions treated with immune modulators. Another limitation is having identified patients with PI using relevant ICD-10 codes until the first COVID-19 visit. It is possible that a patient may be diagnosed with PI for the first time after the initial COVID-19 visit. This may have resulted in the misclassification of patients with PI as patients without PI. There was an insufficient sample size to separately calculate the adjusted odds ratios for severe outcomes for the types of PI. Different PI types affect different components of the immune system and, as a result, have varying symptoms and severity of illness. It is possible that certain types of PIs are associated with more severe COVID-19 outcomes, while others are not. Another potential limitation is selection bias [36] due to the selection of participants from the ED. Only a subset of PI and non-PI patients with COVID-19 were selected. Specifically, this may have introduced collider bias [36], since PI patients may be more likely to visit the hospital upon infection and individuals with more severe COVID-19 may be more likely to go to the ED than patients with asymptomatic or mild COVID-19. All comorbidities contributing to COVID-19 severity could not be accounted for in the model. Another limitation is not having accounted for the choice of treatment among different hospitals. The differential treatments provided by different hospitals could not be easily categorized. As a result, this may have affected the study results.

## 5. Conclusions

The effect of PIs on immune function makes PIs important chronic diseases to understand in relation to infectious disease risk. Furthermore, understanding the pathology of COVID-19 in individuals with PI can provide insights into how to better treat these patients. These findings can inform public health messaging on comorbidities contributing to increased COVID-19 severity to ensure that people with PI and their healthcare providers are aware of the evidence regarding risk of adverse outcomes among these patients.

## Figures and Tables

**Table 1 jcm-12-03516-t001:** ICD-10 code labels and frequencies for ascertaining patients with PI from 1 April 2020 to 31 August 2021.

ICD-10 Code ^a^	Frequency, No. (%)	Label
D83.9	318 (31.2%)	Common variable immunodeficiency, unspecified
D80.3	250 (24.5%)	Selective deficiency of immunoglobulin G (IgG) subclass
D84.1	83 (8.1%)	Defects in the complement system
D80.2	79 (7.7%)	Selective deficiency of immunoglobulin A (IgA)
D80.0	32 (3.1%)	Hereditary hypogammaglobulinemia
D80.4	32 (3.1%)	Selective deficiency of immunoglobulin M (IgM)
D80.9	27 (2.6%)	Immunodeficiency with predominantly antibody defects, unspecified
D82.4	25 (2.5%)	Hyperimmunoglobulin E (IgE) syndrome
D80.8	24 (2.4%)	Other immunodeficiencies with predominantly antibody defects
M04.1	24 (2.4%)	Periodic fever syndromes
D82.1	20 (2.0%)	Di George’s syndrome
D83.8	<20	Other common variable immunodeficiencies
D80.6	<20	Antibody deficiency with near-normal immunoglobulins or with hyperimmunoglobulinemia
D89.82	<20	Autoimmune lymphoproliferative syndrome (ALPS)
D83.0	<20	Common variable immunodeficiency with predominant abnormalities of B-cell numbers and function
M04.8	<20	Other autoinflammatory syndromes
D83.1	<20	Common variable immunodeficiency with predominant immunoregulatory T-cell disorders
D80.5	<20	Immunodeficiency with increased immunoglobulin M (IgM)
D81.818	<20	Other biotin-dependent carboxylase deficiency
D82.3	<20	Immunodeficiency following hereditary defective response to Epstein–Barr virus
D82.9	<20	Immunodeficiency associated with major defect, unspecified
D81.819	<20	Other biotin-dependent carboxylase deficiency, unspecified
D82.0	<20	Wiskott–Aldrich syndrome
D84.0	<20	Lymphocyte function antigen-1 (LFA-1) defect
D82.8	<20	Immunodeficiency associated with other specified major defects
D83.2	<20	Common variable immunodeficiency with autoantibodies to B or T cells
D80.7	0 (0%)	Transient hypogammaglobulinemia of infancy
D81.0	0 (0%)	Severe combined immunodeficiency (SCID) with reticular dysgenesis
D81.1	0 (0%)	SCID with low T- and B-cell numbers
D81.2	0 (0%)	SCID with low or normal B-cell number
D81.3	0 (0%)	Adenosine deaminase (ADA) deficiency
D81.4	0 (0%)	Nezelof’s syndrome
D81.5	0 (0%)	Purine nucleoside phosphorylase (PNP) deficiency
D81.6	0 (0%)	Major histocompatibility complex class I deficiency
D81.7	0 (0%)	Major histocompatibility complex class II deficiency
D81.810	0 (0%)	Biotinidase deficiency
D81.89	0 (0%)	Other combined immunodeficiencies
D81.9	0 (0%)	Combined immunodeficiency, unspecified
D82.2	0 (0%)	Immunodeficiency with short-limbed stature
D83	0 (0%)	Common variable immunodeficiency
M04.2	0 (0%)	Cryopyrin-associated periodic syndromes
Total	1020 (100%)	Any primary immunodeficiency (Any PI)

^a^—Patients may have more than one primary immunodeficiency-related ICD-10 code.

**Table 2 jcm-12-03516-t002:** Patient characteristics in PHD dataset from 1 April 2020 to 31 August 2021.

Characteristic		Primary Immunodeficiency	Non-Primary Immunodeficiency
All, No. (row %)		853 (100%)	1,197,430 (100%)
Gender	Female	548 (64.2%)	627,785 (52.4%)
	Male	305 (35.8%)	569,645 (47.8%)
Age	18–39	144 (16.9%)	305,230 (25.5%)
	40–54	167 (19.6%)	277,375 (23.2%)
	55–64	187 (21.9%)	215,113 (18.0%)
	65–74	185 (21.7%)	188,704 (15.8%)
	75+	170 (19.9%)	211,008 (17.6%)
Race/Ethnicity	Hispanic	75 (8.8%)	257,666 (21.5%)
	Non-Hispanic black	79 (9.3%)	242,556(20.3%)
	Non-Hispanic white	649 (76.1%)	581,790 (48.6%)
	Other	38 (4.5%)	88,541 (7.4%)
	Unknown	<20	26,877 (2.2%)
Comorbidities	Neoplasm	191 (22.4%)	54,043 (4.5%)
	Chronic obstructive pulmonary disease	305 (35.8%)	116,591 (9.7%)
	Chronic kidney disease	190 (22.3%)	142,879 (11.9%)
	Obesity	298 (32.9%)	242,324 (20.2%)
	Type 2 diabetes	308 (36.1%)	324,018(27.1%)
	Cardiovascular disease	388 (45.5%)	281,763 (23.5%)

**Table 3 jcm-12-03516-t003:** Characteristics of included PI and non-PI patients in PHD from 1 April 2020 to 31 August 2021 by COVID-19 outcome.

Characteristic		Primary Immunodeficiency					Non-Primary Immunodeficiency				
		ED Visit	Hospitalization	ICU	IMV	Death	ED Visit	Hospitalization	ICU	IMV	Death
		No. (%)	No. (%)	No. (% ^a^)	No. (% ^a^)	No. (% ^a^)	No. (%)	No. (%)	No. (% ^a^)	No. (% ^a^)	No. (% ^a^)
All, No. (row %)		853 (100%)	624 (73.2%)	183 (29.3%)	121 (19.4%)	111 (17.8%)	1,197,430 (100%)	535,663 (44.7%)	118,566 (22.1%)	72,610 (13.6%)	76,594 (14.3%)
Gender	Female	548 (100%)	389 (71.0%)	112 (28.8%)	67 (17.2%)	59 (15.2%)	627,785 (100%)	254,989 (40.6%)	49,509 (19.4%)	29,139 (11.4%)	32,227 (12.6%)
	Male	305 (100%)	235 (77.1%)	71 (30.2%)	54 (23.0%)	52 (22.1%)	569,645 (100%)	280,674 (49.3%)	69,057 (24.6%)	43,471 (15.5%)	44,367 (15.8%)
Age	18–39	144 (100%)	78 (54.2%)	24 (30.8%)	<20	<20	305,230 (100%)	53,042 (17.4%)	8667 (16.3%)	4009 (7.6%)	1311 (2.5%)
	40–54	167 (100%)	110 (65.9%)	30 (27.3%)	22 (20.0%%)	<20	277,375 (100%)	97,441 (35.1%)	19,241 (19.7%)	11,010 (11.3%)	5693 (5.8%)
	55–64	187 (100%)	142 (75.9%)	41 (28.9%)	25 (17.6%)	23 (16.2%)	215,113 (100%)	109,598 (51.0%)	26,188 (23.9%)	16,951 (15.5%)	11,768 (10.7%)
	65–74	185 (100%)	149 (80.5%)	45 (30.2%)	36 (24.2%)	32 (21.5%)	188,704 (100%)	119,010 (63.1%)	31,541 (26.5%)	21,305 (17.9%)	20,034 (16.8%)
	75+	170 (100%)	145(85.3%)	43 (29.7%)	23 (15.9%)	34 (23.4%) ^b^	211,008 (100%)	156,572 (74.2%)	32,929 (21.0%)	19,335 (12.3%)	37,788 (24.1%)
Race/Ethnicity	Hispanic	75 (100%)	47 (62.7%)	<20	<20	<20	257,666 (100%)	99,548 (38.6%)	22,727 (22.8%)	14,673 (14.7%)	13,236 (13.3%)
	Non-Hispanic black	79 (100%)	54 (68.4%)	<20	<20 ^b^	<20 ^b^	242,556 (100%)	101,304 (41.8%)	22,614 (22.3%)	14,102 (13.98%)	12,990 (12.8%)
	Non-Hispanic white	649 (100%)	486 (74.9%)	136 (28.0%)	94 (19.3%)	90 (18.5%)	581,790 (100%)	282,771 (48.6%)	61,455 (21.7)	35,415 (12.5%)	42,080 (14.9%)
	Other	38 (100%)	27 (71.1%)	<20	<20	<20 ^b^	88,541 (100%)	40,350 (45.6%)	9042 (22.4%)	6538 (16.2%)	6491 (16.1%)
	Unknown	<20 (100%)	<20 (83.3%)	<20 ^b^	<20 ^b^	<20 ^b^	26,877 (100%)	11,690 (43.5%)	2728 (23.3%)	1882 (16.1%)	1797 (15.4%)
Comorbidities	Neoplasm	191 (100%)	153 (80.1%) ^b^	48 (31.4%)	26 (17.0%) ^b^	38 (24.8%) ^b^	54,043 (100%)	41,076 (76.0%)	10,174 (24.8%)	6213 (15.1%)	8460 (20.6%)
	Chronic obstructive pulmonary disease	305 (100%)	246 (80.7%) ^b^	81 (32.9%)	52 (21.1%) ^b^	55 (22.4%) ^b^	116,591 (100%)	89,136 (76.5%)	24,260 (27.2%)	15,447 (17.3%)	18,642 (20.9%)
	Chronic kidney disease	190 (100%)	154 (81.1%) ^b^	61 (39.6%)	40 (26.0%)	46 (29.9%) ^b^	142,879 (100%)	117,518 (82.3%)	32,870 (28.0%)	21,367 (18.2%)	27,587 (23.5%)
	Obesity	298 (100%)	238 (79.9%)	85 (35.7%)	62 (26.1%)	49 (20.6%)	242,324 (100%)	173,657 (71.7%)	46,109 (26.6%)	30,208 (17.4%)	24,594 (14.2%)
	Type 2 diabetes	308 (100%)	243 (78.9%)	88 (36.2%)	62 (25.5%)	58 (23.9%)	324,018 (100%)	223,072 (68.9%)	60,451 (27.1%)	38,428 (17.2%)	38,830 (17.4%)
	Cardiovascular disease	388 (100%)	320 (82.5%)	111 (34.7%)	74 (23.1%)	80 (25.0%) ^b^	281,763 (100%)	214,185 (76.0%)	61,940 (28.9%)	40,506 (18.9%)	49,042 (22.9%)

^a^—Percent of COVID-19 outcome in hospitalizations. ^b^—Non-statistically significant difference between patients with PI and patients without PI.

**Table 4 jcm-12-03516-t004:** Frequency of severe COVID-19 outcomes among top 4 PI groups.

ICD-10 Code	Primary Immunodeficiency	Total	Hospitalized	ICU ^a^	IMV ^a^	Deaths ^a^
D83.9	Common variable immunodeficiency, unspecified	318	216 (67.9%)	71 (32.9%)	52 (24.1%)	44 (20.4%)
D80.3	Selective deficiency of immunoglobulin G (IgG) subclass	250	188 (75.2%)	48 (25.5%)	37 (19.7%)	34 (18.1%)
D84.1	Defects in the complement system	83	55 (66.3%)	23 (41.8%)	<20	<20
D80.2	Selective deficiency of immunoglobulin A (IgA)	79	46 (58.2%)	<20	<20	<20

^a^—Percent out of hospitalized patients.

**Table 5 jcm-12-03516-t005:** Odds ratios for COVID-19 outcomes between PI and non-PI patients.

1 April 2020–31 August 2021			
COVID-19 Outcome	Non-Primary Immunodeficiency Patients (*n* = 1,197,430)	Primary Immunodeficiency Patients (*n* = 853)	Adjusted Odds Ratio ^a^ (95% CI)
Hospitalization	535,663 (44.7%)	624 (73.2%)	2.36 (1.87–2.98)
ICU admission	118,566 (9.9%)	183 (21.5%)	1.53 (1.19–1.96)
Invasive mechanical ventilation	72,610 (6.1%)	121 (14.2%)	1.41 (1.15–1.72)
Death	76,594 (6.4%)	111 (13.0%)	1.37 (1.08–1.74)

^a^—These odds ratios were adjusted for age, sex, race/ethnicity, cardiovascular disease, type 2 diabetes, chronic obstructive pulmonary disease, chronic kidney disease, cancer, and obesity.

## Data Availability

Restrictions apply to the availability of these data. Data were obtained from PINC-AI^TM^ Applied Sciences of Premier.

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
