# Peer review of "Severity Outcomes among Adult Patients with Primary Immunodeficiency and COVID-19 Seen in Emergency Departments, United States, April 2020–August 2021"

_jcm, 2023, doi:10.3390/jcm12103516_

Round 1
Reviewer 1 Report
In this study, the Authors analyzed COVID-19 outcome in a large cohort of Primary Immunodeficiency patients in comparison with non-PI patients. The study is noteworthy. However, a few points need to be further clarified.
1. The Author did not analyzed the outcome according to the severity of the immunological disease. For instance, pts with CVI were grouped along with IgA deficiency. A differential analysis according to the severity of the PI will add to the value of the study.
2. Pts with PI, and in particular those with humoral defects, are usually hospitalized more often than the general population. Thus, the number of hospitalizations in PI should also take in consideration the rate of hospitalization per year before Covid.
Author Response
We appreciate the opportunity to revise and resubmit. Addressing the astute comments from the reviewer has allowed us to clarify our methods and results and generally improve our message.
Reviewer 1
1) The Author did not analyzed the outcome according to the severity of the immunological disease. For instance, pts with CVI were grouped along with IgA deficiency. A differential analysis according to the severity of the PI will add to the value of the study.
We appreciate this comment. We agree that differential analysis within different PI groups would add value to this study. However, we were unable to perform these analyses due to the small sample sizes in these groups. We have stated this in the limitations (lines 229-232).
“There was an insufficient sample size to calculate adjusted odds ratios for severe outcomes for the types of PI separately. Different PI types affect different components of the immune system and as a result have varying symptoms and severity of illness. It is possible that certain types of PI are associated with more severe COVID-19 outcomes while others are not.”
2) Pts with PI, and in particular those with humoral defects, are usually hospitalized more often than the general population. Thus, the number of hospitalizations in PI should also take in consideration the rate of hospitalization per year before Covid.
Thank you for your comment. The goal of our study was to confirm our hypothesis that PI is a risk factor for adverse outcomes in COVID patients. And so, while baseline rates of hospitalization rates for PI patients before COVID are important, the most appropriate comparison would be to others serious respiratory diseases such as influenza and we respectfully submit that extending the analyses to analyzing adverse outcomes such as hospitalization among PI and non-PI patients with influenza would be out of scope for this study.
Reviewer 2 Report
The authors attempt to correlate primary immunodeficiency with the impact of covid-19 in a large group of patients and summarize their results in Table 4 where they indicate that there is a higher number of patients with PI in the most critical conditions compared to non-PI. The work has the merit of being the largest study correlating both pathologies.
In my opinion, I believe that the authors should verify whether the results obtained are significantly different in the populations studied.
for example in the paragraph:
"After being seen in the ED, from April 1, 2020 to August 31, 2021, 73.2% of PI patients and 44.7% of non-PI patients were hospitalized, 29.3% of hospitalized PI patients and 22.1% of hospitalized non-PI patients were admitted to the ICU, 19.4% of hospitalized PI patients and 13.6% of hospitalized non-PI patients had IMV due to COVID-19, and 17.8% of hospitalized PI patients and 14.3% of non-PI hospitalized patients died"
Do the % values shown in this text present statistical significative differences (SSD)?
In the group of ICD-10 codes, of those with more than 2%, which was the most affected due to COVID-19?
The authors do not mention throughout the manuscript what was the length of stay at the hospital of PI patients vs non-PI patients, could you please include this data if available, and show if there is SSD?
The choice of treatment in the different hospitals may affect the results shown in this study, how did you controlled that?
There are studies that compare the PI in different ethnic groups that can help explain the differences observed in this study beyond the low representativeness of the groups studied.
The conclusions are weak, you have many results that you do not exploit. On the other hand, they do not mention the form of treatment in each group and between the different hospitals, in my opinion this point is critical and its analysis may affect the results found.
Author Response
We appreciate the opportunity to revise and resubmit. Addressing the astute comments from the reviewer has allowed us to clarify our methods and results and generally improve our message.
Reviewer 2
1) Do the % values shown in this text present statistical significative differences (SSD)?
Thank you for pointing this out. The values with statistically significant differences have been denoted in the table 3.
2) In the group of ICD-10 codes, of those with more than 2%, which was the most affected due to COVID-19?
We agree that differential analysis within different PI groups would add value to this study. We added the frequency of COVID-19 outcomes for the top 4 PI subgroups (lines 158-163, table 4).
“Of the largest 4 PI groups (common variable immunodeficiency, selective deficiency of immunoglobulin G [IgG] subclass, defects in the complement system, and selective deficiency of immunoglobulin A [IgA]), selective deficiency of immunoglobulin G [IgG] subclass has the highest frequency of hospitalization (75.2%) and defects in the complement system had the highest frequency of ICU admission (41.8%), IMV (30.9%), and deaths (21.8%) of hospitalized patients (table 4).”
3) The authors do not mention throughout the manuscript what was the length of stay at the hospital of PI patients vs non-PI patients, could you please include this data if available, and show if there is SSD?
We agree that length of stay for PI vs non-PI patients would be important to add to the data. The difference in length of hospital stay has been added the results (line 166-167).
“PI patients spent on average 1.91 more days (p < 0.001) in the hospital than non-PI patients after adjusting for the covariates.”
4) The choice of treatment in the different hospitals may affect the results shown in this study, how did you controlled that?
We agree that treatment choice in different hospitals may affects the results. We accounted for correlation among patients within the same hospital, but we cannot easily categorize the differential treatments/ healthcare provided by different hospitals. We included this as a limitation of the study. (lines 238-240)
“Another limitation is not accounting for the choice of treatment between different hospitals. The differential treatments could not be easily categorized provided by different hospitals. As a result, this may affect the study results.”
5) There are studies that compare the PI in different ethnic groups that can help explain the differences observed in this study beyond the low representativeness of the groups studied.
Thank you for pointing this out. We agree that it is important to acknowledge the disparities in prevalence of diagnosed PIs in different ethnic groups. We have mentioned this to the discussion (lines 212-215).
“However, it is important to point out that the majority of PI patients (76.1%) were non-Hispanic White while 9.3% were non-Hispanic Black and 8.8% were Hispanic. The underrepresentation of minority populations is likely due to an overall disparity in PI diagnosis that may be due to differences in patient referral and access to care between these groups[32-35].”
6) The conclusions are weak, you have many results that you do not exploit. On the other hand, they do not mention the form of treatment in each group and between the different hospitals, in my opinion this point is critical and its analysis may affect the results found.
We agree that the form of treatment may vary between hospitals as well as between patients within the same hospital depending on the severity of condition. However, the focus of our study was on severe outcomes of COVID-19 (hospitalization, ICU stay, IMV use, and deaths). The treatment for a patient in ICU or on IMV may be different from other patients in a hospital. We respectfully submit that these variations with regard to the treatment and management of PI patients is not easily or uniformly available in the study database and as such, we are not able to control of this this. We have noted this in the discussion and limitations (lines 238-240).
“Another limitation is not accounting for the choice of treatment between different hospitals. The differential treatments could not be easily categorized provided by different hospitals. As a result, this may affect the study results.”
Reviewer 3 Report
In this manuscript, authors investigated severity outcomes of COVID-19 infections in adult PI patients, compared to non-PI patients.
The strength of the manuscript is the large sample size, and all potential limitations have been clearly defined, including the use of ICD-10 codes to identify PI patients.
On this point, majority of PI patients presented antibody deficiencies, and ther only a small percentage of autoimmune or dysregulation defects, and no T-cell defects. However, it is logical when considering only adult patients.
I have some minor comments:
- why stopping at first COVID-19 visit when looking at ICD-10 codes to identify PI and associated comorbities? I suggest to search for these codes on all the studied period, as COVID-19 infection could eventually be the first presentation of PI disease?
- when considering the official classification of PI (ref. 4), I'm not sure about including biotin-dependant carboxylase deficiency and biotinidase deficiency. However, their number should not impact global outcome.
- when comparing to previous studies, could you precise if those studies are looking at adult or children PI?
- Finally, I suggest to update ref.4 to the last publication (doi: 10.1007/s10875-022-01289-3)
Author Response
We appreciate the opportunity to revise and resubmit. Addressing the astute comments from the reviewer has allowed us to clarify our methods and results and generally improve our message.
Reviewer 3
1) why stopping at first COVID-19 visit when looking at ICD-10 codes to identify PI and associated comorbities? I suggest to search for these codes on all the studied period, as COVID-19 infection could eventually be the first presentation of PI disease?
Thank you for pointing this out. We agree that it is possible to miss undiagnosed PI cases following a COVID-19 diagnosis. We have added this to the limitations (lines 226-229).
“Another limitation is identifying patients with PI through relevant ICD-10 codes until the first COVID-19 visit. It is possible that the patient may be diagnosed with PI for the first time after the initial COVID-19 visit. This may result in misclassification of patients with PI to patients without PI”
2) when considering the official classification of PI (ref. 4), I'm not sure about including biotin-dependant carboxylase deficiency and biotinidase deficiency. However, their number should not impact global outcome.
Biotin-dependant carboxylase deficiency and biotinidase deficiency are on the official list from the American academy of allergy asthma and immunology and we would like to include it for completeness. Yes, we agree that it would not affect global impact as these diagnoses are extremely rare.
3) when comparing to previous studies, could you precise if those studies are looking at adult or children PI?
Thank you for pointing this out. The studies are looking at adult, children, or both has been specified. (lines 179-182, 188-190).
“A study in the United Kingdom found that 53.3% of 60 adult PI and COVID-19 cases were hospitalized, the case fatality ratio was 31.6%, and the inpatient mortality ratio was 37.5%[11]. Another study of 16 adult PI patients positive for SARS-CoV-2 in the United States found the case fatality rate to be 25%[7].
“Concerning COVID-19 infection, Milito (2021) found a similar incidence in Italy between all PI patients (4.01 per 100,000) and the general population (5.22 per 100,000)[17]. Delavari (2021) found children PI patients in Iran to have a 1.23 higher risk of COVID-19 infection than the general population[19].”
4) Finally, I suggest to update ref.4 to the last publication (doi: 10.1007/s10875-022-01289-3)
Thank you for pointing this out. The current ref. 4 is not the most updated publication, however, this was the version that was used when determining PI related ICD-10 codes.
Reviewer 4 Report
This is a great report. It is best to add the expression changes of some cytokines, chemokines, complement proteins, etc. to this article. It is also best to add some healing results to this article
This is a great report. It is best to add the expression changes of some cytokines, chemokines, complement proteins, IgG etc. to this article. It is also best to add some healing results to this article
Author Response
We appreciate the opportunity to revise and resubmit. Addressing the astute comments from the reviewer has allowed us to clarify our methods and results and generally improve our message.
Reviewer 4
1) This is a great report. It is best to add the expression changes of some cytokines, chemokines, complement proteins, etc. to this article. It is also best to add some healing results to this article
Thank you. We agree that expression changes of cytokines, chemokines, and complement proteins and healing results are important to study, but the focus of this paper is severity of Covid-19 for PI patients compared to non-PI patients. We are wondering if the reviewer is referring to treatment/management of PI patients as “Healing results”. In that sense, we respectfully submit that data on specific treatment and management of PI patients is not easily or uniformly available in the study database and as such, we are not able to add these to the results. We have noted this in the discussion and limitations. We did add the difference between length of hospital stay between PI patients and non-PI patients (line 166-167).
“PI patients spent on average 1.91 more days (p < 0.001) in the hospital than non-PI patients after adjusting for the covariates.”
Round 2
Reviewer 2 Report
Dear Dr. Drzymalla, I agree with all your answers, however now the article is stronger and needs to improve its abstract and conclusion. In my opinion you have to catch the readers by including more data in the abstract. Now you have included more information, for example Table 4 summarizes the ICD-10 codes with the highest frequency of hospitalization, ICU, IMV and death, however you did not include part of that information in the abstract or in the conclusion, that is, again In my opinion, a very important fact. The same comment is valid for tha data of hospitalization time, 1.91 more days in the case of PI vs non-PI is an important reduction (based on your p<0,001)
Author Response
Reviewer comments
Thank you for your comment and for the ability to improve our abstract.
Reviewer 2
I agree with all your answers, however now the article is stronger and needs to improve its abstract and conclusion. In my opinion you have to catch the readers by including more data in the abstract. Now you have included more information, for example Table 4 summarizes the ICD-10 codes with the highest frequency of hospitalization, ICU, IMV and death, however you did not include part of that information in the abstract or in the conclusion, that is, again In my opinion, a very important fact. The same comment is valid for tha data of hospitalization time, 1.91 more days in the case of PI vs non-PI is an important reduction (based on your p<0,001)
Thank you for pointing this out. We agree that this information is important to add in the abstract. We have added the information for the hospitalization time and the summary data in table 4 to the abstract. (lines 44-47).
“Hospitalization, intensive care unit (ICU) admission, invasive mechanical ventilation (IMV), and death had higher odds for PI patients than non-PI patients (Hospitalization aOR: 2.36 95% CI: 1.87-2.98, ICU admission aOR: 1.53, 95% CI: 1.19-1.96, IMV aOR: 1.41, 95% CI: 1.15-1.72, Death aOR: 1.37, 95% CI: 1.08-1.74) and PI patients spent on average 1.91 more days in the hospital than non-PI patients when adjusted for age, sex, race/ethnicity, and chronic conditions associated with severe COVID-19. Of the largest 4 PI groups, selective deficiency of immunoglobulin G subclass has the highest hospitalization frequency (75.2%).”